# Towards the Use of Big Data in Healthcare: A Literature Review

**DOI:** 10.3390/healthcare10071232

**Published:** 2022-07-01

**Authors:** Grazia Dicuonzo, Graziana Galeone, Matilda Shini, Antonella Massari

**Affiliations:** Department of Economics, Management and Business Law, University of Bari Aldo Moro, Largo Abbazia Santa Scolastica, 53, 70124 Bari, Italy; grazia.dicuonzo@uniba.it (G.D.); matilda.shini@uniba.it (M.S.); antonella.massari@uniba.it (A.M.)

**Keywords:** artificial intelligence, big data analytics, healthcare

## Abstract

The interest in new and more advanced technological solutions is paving the way for the diffusion of innovative and revolutionary applications in healthcare organizations. The application of an artificial intelligence system to medical research has the potential to move toward highly advanced e-Health. This analysis aims to explore the main areas of application of big data in healthcare, as well as the restructuring of the technological infrastructure and the integration of traditional data analytical tools and techniques with an elaborate computational technology that is able to enhance and extract useful information for decision-making. We conducted a literature review using the Scopus database over the period 2010–2020. The article selection process involved five steps: the planning and identification of studies, the evaluation of articles, the extraction of results, the summary, and the dissemination of the audit results. We included 93 documents. Our results suggest that effective and patient-centered care cannot disregard the acquisition, management, and analysis of a huge volume and variety of health data. In this way, an immediate and more effective diagnosis could be possible while maximizing healthcare resources. Deriving the benefits associated with digitization and technological innovation, however, requires the restructuring of traditional operational and strategic processes, and the acquisition of new skills.

## 1. Introduction

The adoption of Fourth Industrial Revolution technologies, particularly artificial intelligence (AI) and big data (BD), has been a major challenge for all industries [1]. The increasing technological progress has initiated a digital transformation process in many sectors, including healthcare [2], which is already moving toward Healthcare 4.0 due to the impact of smart technologies [3,4] such as the Internet of Things (IoT) paradigm [5], cloud and fog computing [5], and big data analytics (BDA) [6].

Healthcare institutions often face many challenges, ranging from epidemics to determining the most suitable therapies for treating diseases. If an AI technology system is applied to medical research, owing to the development, validation, and deployment of various machine learning algorithms for industrial applications with sustainable performance [7], it has the potential to diagnose, find vaccines, and personalize healthcare services, moving toward highly advanced e-Health [8].

Patient-centered care cannot ignore the continuous expansion of data in terms of its volume, variety, and velocity, propelling it toward a new technological paradigm, now widely called BD [9,10]. The analysis of the enormous volume, heterogeneity, and velocity of the information provided by BD allows for the extraction of the greatest value from collected data and successfully solving and analyzing the relationships between different variables that describe a patient’s vital functions and that can affect their health [11]. These data stimulate healthcare organizations to invest heavily in data analysis to facilitate decision-making [12,13]. Integrating data on an individual’s unique characteristics, clinical phenotypes, and biological information obtained from diagnostic imaging to laboratory tests and medical records enables precision medicine to operate under predictive and preventive conditions [14]. Having abundant data is crucial, especially in critical care environments, to be able to rapidly identify diagnoses and specific treatments for particular or rare pathological cases [15]. The improvement of critical stages of diagnosis and the personalization of therapeutic treatments for various diseases are spreading rapidly due to the emerging technological development of BD and the use of social media and IoT that allow for collecting various kinds of data generated by a huge number of devices. In particular, these are biomedical sensors and intelligent devices that, during the diagnosis and monitoring of a patient, collect data related to their health and make them accessible through interconnected and integrated systems, facilitating the transmission of information [5,16].

Currently, the health emergency situation caused by the spread of the COVID-19 disease is increasing the need to develop a BD information system for epidemic- and rapid problem-oriented BD acquisition and integration [17,18]. Previous studies examining BD in healthcare predominantly focused on informatics [16,19,20] and medical aspects [21,22]. Few studies have analyzed this topic from a managerial point of view [23,24]. Therefore, this paper contributes to this stream of literature by exploring which tangible and intangible elements are needed to draw the maximum benefits from BD.

Scopus has been used to select the most relevant studies on the role of BD in healthcare to generate knowledge even from the most remote contributions to the literature [25]. The study shows exponential growth in publications, especially since 2020, highlighting the growing criticality and urgency of healthcare and Industry 4.0 integration. Given the increasing interest in BD in healthcare management, a literature review is useful to understand the challenges and opportunities of BD’s use [22] for future applications in healthcare.

## 2. Background

BD is a large collection of data from various healthcare sources that enables increasingly personalized treatments, evaluations of their effectiveness, and a reduction of clinical risk through innovative ways of managing and controlling processes [26,27,28].

The literature identifies the main features that characterize BD, also known as the 3Vs [28,29]:Volume: amount of data generated every second;Variety: different types of data generated, accumulated, and used, even unstructured or semi-structured; andVelocity: referring to the generation of data (which is always increasing).

To these first 3Vs, another 3Vs were later added [30,31]:Veracity or uncertainty of data;Value: BD analytics technologies increase the value of data by transforming it into useful information; andVariability: data on the same topic can have differences related to their format or mode of collection, and this is often a limitation.

Then a seventh additional characteristic was found:Complexity: the larger the size of the dataset, the greater the complexity of the data to be managed.

Although the importance of BD has emerged particularly in finance, banking, and insurance, one of the most promising and interesting areas in which it can effect significant change is healthcare, although its adoption has been slow [32,33]. Global BD in the healthcare market was expected to grow at a CAGR (Compounded Average Growth Rate) of 20.69% between 2015 and 2021 [34]. BD is transforming the way healthcare is managed, enabling a revolution in knowledge management and data analytics [35]. The analysis of the large amount of data generated by a single patient related to diagnosis, treatment pathways, drugs, medical devices, digital images, and laboratory analysis results, to be meaningful, requires that these data be validated, processed, and integrated into processing systems that allow for creating new value in the organization of health services [36]. The data, therefore, are stored in databases and are efficiently managed [9,37], providing useful insights that otherwise would not have been possible and identifying better solutions in terms of health quality and timely decisions [23,38].

The increasing influence of BD has prompted healthcare organizations to use AI and the skills needed to effectively exploit BDA [1,24]. The quantity of data to manage, analyze, and archive is so large and complex that traditional methods of data processing are inadequate [39]. The potential acquisition and analysis of BD, in fact, requires the restructuring of the technological infrastructure and integration of traditional data analytical tools and techniques with computational technology that is able to enhance and extract information that will be useful for decision-making [2,36,40].

The most relevant sources from which to acquire BD in healthcare are medical recordings (e.g., electronic healthcare records, clinical decision support systems, biomedical data, etc.) [41] as well as external data sources (laboratories, pharmacies, patient-reported data, biometric and other data received directly from patients, etc.). Additional data sources are increasingly available, such us data derived from Internet use (social media) and smart applications [42,43,44,45]. For the management and processing of these data, many healthcare sectors have adopted cloud computing. It is a solution for receiving and storing huge amounts of patient data and managing electronic medical records [46]. These heterogeneous data, when properly integrated with the most relevant health data, allow for the monitoring of patients’ health status in various contexts (hospitals, nursing homes, private homes) [5,16,47]. This is an aspect of crucial relevance because the main errors that can lead to a misdiagnosis and case fatality occur due to improper monitoring and administration of therapeutic treatment, as well as to drug non-adherence [48]. Integrating data on an individual’s unique characteristics, clinical phenotypes, and biological information obtained from imaging to laboratory tests and medical records enables individualized diagnostic or therapeutic solutions [14].

However, there are still many challenges to be faced. The use of cloud computing and other BD analysis tools and techniques in general, in fact, encounters a number of difficulties represented by network failures, security and privacy issues of patient data, and network downtime [46,49,50].

The proliferation of increasingly fast network infrastructures is a phenomenon that is directly proportional to the expansion of possibilities of conveying and exchanging health information, opening up scenarios that were unimaginable until a few years ago. However, today, medicine and scientific research in the medical field are no longer carried out with traditional devices but also, for example, through the so-called smart devices that are increasingly becoming essential elements of daily life [51].

A product of the technological revolution that began prior to 2000 with the explosion of the Internet, and later with the huge spread of new generation devices connected to it (IoT), is e-Health. In fact, e-Health has huge potential for improving the efficiency of the health system (cutting costs) and effectiveness in the management of patients (understood as the quality of healthcare) [52,53]. The convergence toward a health system, Healthcare 4.0 [3,4] based on smart technologies, IoT [5], data sharing between different actors [6], robotics, and cloud computing [54] can lead to improved healthcare delivery. These IoT devices and sensors also play an essential role in analyzing and predicting new diseases, such as COVID-19 [55].

The affirmation of new technologies has determined the creation of the Digital Imaging and Communication in Medicine (DICOM) standard, which defines the rules for the storage and sharing of images, going beyond the old generation of analog machines. Another AI application relates to digital reporting techniques—that is, electronic health records (EHR), which in a few years will replace paper media [56]. To protect patient privacy, EHR must be stored as sensitive information in a secure and reliable manner [57].

The main health benefits of BD are found in disease prevention, in identifying the main health risk factors, and in designing more effective healthcare measures [15,58,59]. The rational use of information and communications technology (ICT) represents a revitalization lever for health systems challenged especially during the COVID-19 health emergency. Enhancing decision-making and operational capabilities, reducing errors, and saving resources are the key benefits. In this view, BD is proving to be an important source with new characteristics, potential, and limitations [60,61].

BD and AI technologies have high predictive capabilities in their application in the treatment of cardiovascular diseases. Studies [62,63] have shown that it is mainly machine learning techniques (k-nearest neighbor, decision tree, linear regression, and support vector machine [SVM]) that improve the accuracy of heart disease detection.

Combinations of machine learning methods with deep learning approaches also enhance the use of neuroimaging data to classify and predict Alzheimer’s disease [64]. Alzheimer’s is a degenerative neurological disease that impairs a person’s ability to function independently, making early diagnosis critical. Sharma et al. proposed a Hadoop-based BD system for early indicators of the disease. Such a system involves combining data obtained from noninvasive magnetic resonance imaging (MRI), spectrography, magnetic resonance spectroscopy (MRS), and neuropsychological test results [65]. Kautzky et al. (2018), however, developed a prediction model based on a single diagnostic factor that allows early detection of brain abnormalities even before the onset of symptoms [66].

An additional degenerative disease in which to employ machine learning is Parkinson’s disease, which affects the neurological system and limits mobility. For early detection of disease symptoms, some studies have used k-nearest neighbor, random forest, and decision tree algorithms [67,68]. Sivaparthipan et al. (2019) also highlighted the importance of data collection using cell phones to recognize the gait of Parkinson’s patients [69].

Different machine learning (ML) techniques are also used to improve the prediction results for cancer, a major cause of mortality globally [70]. Torkey et al. (2021) proposed two survival prediction models based on deep learning that can guide physicians in determining breast cancer treatment options and avoid ineffective treatments [71]. In another study, Torkey et al. (2021) used an ML model that, through the construction of a DNA microarray dataset, allows for the identification of discriminative features that influence the classification of different kinds of cancer and facilitate their early diagnosis [72].

## 3. Materials and Methods

A systematic literature review was conducted over the period 2010–2021 to explore the main areas of application of BD in healthcare and the organizational changes needed to address the challenges of applying BD in this area, as well as to illustrate the potential benefits in light of the COVID-19 health emergency that, with its extemporaneity and unpredictability, has severely affected the healthcare management [73]. A review was also conducted of works from 2022, considering that the scientific production on the investigated topic is already significant.

To ensure a transparent and high-quality process, the analysis comprised four phases [68]:Planning and identification of studiesArticle evaluationExtraction of resultsSummary and dissemination of audit results

The analysis was carried out using the Scopus database [74], and the articles were selected by searching for both “Big Data” and “Healthcare” in the title, abstract, or keywords of an article.

The research conducted, without any restriction on the type of contribution, was delimited with respect to the year of publication and the research area of business, management, and accounting. Subsequently, screening was carried out to assess suitability with respect to the inclusion criteria, first analyzing the relevance of the title, abstract, and then the full text of an article [36]. Works that were not directly related to the definition, process, and use of BD in healthcare management were excluded. Finally, the remaining 93 papers met all inclusion criteria.

Table 1 shows the steps followed in the search strategy:

Table 2 shows the journals that have published the most articles. In particular, International Journal Recent Technology and Engineering presents the highest number of publications (25), covering articles in the areas of computer science and engineering; information technology; electrical and electronics engineering; telecommunication; mechanical, civil, and textile engineering; and all interdisciplinary streams of engineering sciences. This is followed by International Journal of Scientific & Technology Research with 11 publications and Lecture Notes in Business Information Processing with 10 papers in the fields of engineering, science, technology, and industrial application software development. There are many other journals with one paper each.

Applications of BDA in healthcare are gradually increasing with the growing volume of BD in this context since 2014, with new research areas evolving and applications being explored (Figure 1).

## 4. Results

The recent literature regarding BD in healthcare discusses the following three themes:-BD and health awareness-BD and digital transformation-BD and analytical skills

This section presents the results of the analysis, highlighting the benefits, challenges, and risks of BD’s use in the healthcare sector.

### 4.1. Big Data and Health Awareness

The main health benefits of BD are found in disease prevention, in identifying the main health risk factors and in designing more effective healthcare measures [58,59,75]. BD, in fact, supports the digitization of all medical records by making all data related to each patient’s medical history available [76].

The areas that could gain more than others from the benefits of technology and, in particular, from the preservation and sharing of large amounts of clinical data are predictivity, timely diagnosis, and personalized treatment, also favoring the development of precision medicine (patient-centered care) [38,39]. Taking advantage of new biotechnological discoveries allows for going beyond the traditional concept of a “standard patient” to treating the “individual” in their uniqueness [77,78], owing to the analysis of interactions between the different variables that describe the patient’s vital functions and that may affect their health [11], with enormous benefits for medical functions [72,73,74,75,76,77,78,79]. This improves efficiency and is more patient-centric, yielding consistent, suitable, safe, and flexible solutions [39,80] due to the vigorous analysis by applying various machine learning techniques [2].

The predictive use of BDA tools allows, especially in cases of health emergencies, for the prompt reporting of high-risk patients and ensuring more effective and efficient care, thus improving overall healthcare outcomes [9,81,82]. In fact, the heterogeneity, volume, and velocity of the data contribute to the monitoring of population flows and trends, which are of crucial importance both for early diagnoses and for personalized healthcare services [8,12,13]. El Samad et al. (2021), in this regard, conducted a study showing that BD management is considered a key prerogative for the quality of medical services and conditions [83].

AI-based diagnosis systems and algorithms to detect new outbreaks are just some of the tools that could limit the spread of the SARS-CoV-2 virus and related disease COVID-19, thus maximizing healthcare resources [13,17,84] and contributing to the containment of pandemic risk on national territory [85,86]. Abdel-Basset et al. (2021) demonstrated the relevant role that disruptive technologies for COVID-19 analysis, such as AI, Industry 4.0, IoT, Internet of Medical Things (IoMT), BD, virtual reality (VR), drone technology and autonomous robots, 5G, and blockchain, have played in limiting the spread of COVID-19 outbreaks [87]. Another technology that is being widely deployed is Wireless Body Area Networks (WBANs). This is an innovative solution that can restructure healthcare and make pervasive support available to patients [87,88].

The evolution of digital healthcare into mobile healthcare (mobile health) has also made it possible to manage information via apps that a patient can download directly to their smartphone or tablet [89], allowing them to monitor their health status independently and share it with their doctor [44,90]. Therefore, IoT allows a much better and timely diagnosis of the patient’s status and offers medical services via telemedicine, even in remote locations [58,59,60,61,62,63,64,65,66,67,68,69,70,71,72,73,74,75,76,77,78,79,80,81,82,83,84,85,86,87,88,89,90,91] still underdeveloped, where the number of specialists in health services is insufficient [92]. The use of telemedicine is of fundamental importance in this SARS-CoV-2 pandemic period for preventing and managing COVID-19 infection [93].

The many applications of medical IoT in the field of monitoring include sensors to monitor vital parameters, such as blood pressure, heart rate, etc.; smart tags (i.e., chips inserted in clothing) with monitoring and data scanning functions; bracelets or other wearable devices capable of detecting vital parameters and forwarding emergency calls in case of anomalies; and Real-Time Location System (RTLS) (i.e., Global Positioning System [GPS]), which is a satellite-based navigation system used to locate ambulances, patients, doctors, etc. [5,9].

Most of these systems are set for patients suffering from dementia [94] or diabetes [95] and with vascular pathologies [96].

The combination and wise and competent use of these BD sources can help health operators undertaking various individual or collective activities, summarized in precision medicine, predictive medicine, and prevention.

Fanta et al. (2021) argued that digital technologies are also a supporting tool in the healthcare sector’s transition to a circular economy. Indeed, such tools, and IoT in particular, can support the collection of end-of-life healthcare products as well as their recycling, regeneration, and disposal [97,98].

Table 3 shows top ten articles per citations on theme “Big Data and awareness”.

### 4.2. Big Data and Digital Transformation

The literature review showed that the digital health phenomenon is a true paradigm of innovation that allows for increasing the quality of health services and shaping them according to the needs of the patient, proceeding to the control of their health in real time regardless of geographical location [99,100]. It also requires the main political, legal, and medical players to reconsider the risks associated with the processing of data in the health sector; promote a cultural change, even more than an organizational one, in digital transformation; and investigate new protocols for a more efficient and secure transmission of sensitive health data [97,98,99,100,101].

To manage data in a structured way and address the privacy system effectively and more robustly, healthcare organizations are looking for AI and analytics techniques that will enable them to consolidate organizational resources and develop new data-driven and integrated governance [102]. Managing an integrated healthcare solution requires security of medical data, which can be achieved with the cryptosystem, which has been found to be highly secure against attacks and interference [103]. The goal is to allow the development of ways to monitor the health conditions of the population using a huge volume, variety, and velocity of data from a wide range of healthcare networks in an aggregate and anonymous form [104,105,106] and improve its performance.

The optimal use of resources has a critically important role for healthcare operators in assessing the quality of the healthcare service provided and also requires appropriate technologies to ensure the rational use of resources [107,108,109]. Benzidia et al. (2021) claim that extracting new insights from existing volumes of structured and unstructured data related to medical treatments and products improves decision-making and enables a better understanding of each patient’s costs [93,110].

The result may be an important analytical capability of BD through the definition of previously unobserved patterns and improved resource efficiency through the identification of costly healthcare services, such as unnecessary diagnostic tests and additional treatments [19,111].

Introducing advanced digital solutions to explore huge amounts of heterogeneous and unstructured data requires the design of a clear and integrated strategy across all areas of innovation. It is appropriate to start with understanding the actual level of digital maturity to explore its potential benefits driven by BD analytics and to create value for their healthcare organizations [112,113].

Ultimately, healthcare organizations must begin to develop a concrete analysis of how to apply emerging technologies to methods such as diagnostic procedures, treatment protocol development, patient monitoring, drug development, patient diagnose, and epidemic forecasting [45,46,47,48,49,50,51]. In this way, risks can be minimized and decisions can be made from the perspective of improved effectiveness and efficiency [114,115].

To accelerate digitalization, hospitals must invest in technology to automate processes and streamline operations, moving in two distinct directions: focusing on the organizational level (moving from episodic to coordinated care), where telemedicine is prioritized, and introducing digital solutions to enable new models of care (progressing toward personalized care and increasing the focus on prevention and wellness) [93,94,95,96,97,98,99,100,101,102,103,104,105,106,107,108,109,110,111,112,113,114,115,116].

Implementing a Healthcare 4.0 system requires the careful consideration of process improvement as part of the overall plan to achieve maximum benefits from technology adoption [117]. In the absence of a strategy that indicates a precise and coherent direction for the evolution of organizational and technological models, it is easy to get lost within an ever-increasing range of technologies, and perhaps end up choosing and introducing advanced digital solutions in an organizational context that is unable to grasp all the advantages to transform the competitive landscape and improve organizational performance [52]. New technologies enable the creation of high-quality datasets and extract value from them [118,119] but with a rethinking of existing business models [120,121].

Table 4 shows top ten articles per citations on theme “Big Data and digital transformation”.

### 4.3. Big Data and Analytical Skills

BD can offer a major competitive advantage for healthcare providers, especially with regard to reducing therapeutic mistakes by analyzing patient data [120,122]. In fact, software that uses BD efficiently can flag any inconsistencies between a patient’s medical history and their drug allergies with the medications they are taking, thus alerting the referring physician to discontinue therapy. BD can also identify chronically ill citizens, providing them with preventive actions to avoid clogging emergency channels, such as the emergency room [117,118,123].

To seize these important opportunities, however, it is necessary to make a considerable investment in healthcare organizations, for example, in the hiring of analytics experts—that is, professionals capable of identifying problems from the data and proposing the most appropriate solutions [23,124,125]. Gravili et al. (2021) highlighted the crucial role of intangible elements, especially of the intellectual capital in the health sector. The dissemination of new knowledge and specialized skills promotes the sharing of best practices in the health sector. The result is a reduction in mortality rates, better outcomes in terms of cost minimization, and a reduction in hospitalization periods [115,126,127]. De Mauro et al. (2018) proposed a classification of job roles and skill sets needed in the BD and AI era. This classification provides valuable support for business leaders and human resource managers in the selection process and in developing the skills needed to make the most of BD [72,128].

The management of a highly variable amount of data in real time requires not only new tools and methods but also the development of new knowledge and skills that are essential for converting data into a strategic resource and for implementing new management practices or a new organizational culture across the entire organization. A lack of data analytics skills among existing employees may increase data entry errors that could result in placing information in the wrong record, losing valuable information, and limiting the value a business can derive from the data that it captures [129]. BD analysis is essential in defining the patient diagnosis. Therefore, doctors and nurses’ understanding of data undoubtedly has a positive impact on the rapid recovery of patients in hospitals [62,63]. These are professionals with technical skills and multidisciplinary knowledge who can manage a huge volume of data and extract useful information to ensure adequate social and healthcare and support the restructuring of healthcare processes [64,65].

Furthermore, process innovation and efficient scheduling are key to addressing bottlenecks in healthcare management [64,119].

Table 5 shows top ten articles per citations on theme 3 “Big Data and analytical skills”.

## 5. Discussion and Conclusions

In recent years, there has been a process of digitalization and technological innovation in the healthcare sector to enable the transformation of a huge volume of data into valuable health BD, optimize resources, and improve both the patient experience and organizational performance [66]. The main sources of health data are EHR [38], medical data [130], laboratory information systems [131,132,133,134,135], biometric sources, patient-reported data, and social media (wearable devices and sensors that provide information about a patient’s lifestyle) [22,130].

The rapid deployment of new emergency devices (i.e., wireless communications, mobile computing, and mobile devices) and patient monitoring systems has allowed for the focus to be on the design and delivery of digital health services that, leveraging real-time data, foster integrated and effective governance. It is essential to ensure a personalized health service, early disease diagnosis, and support for patient undergoing online care treatments [132]. The gradual implementation of advanced digital solutions will support the clinical team’s decisions and release time for the most value-added clinical activities and treatment of the most complex cases. BD and AI not only have great potential in the fight against infectious diseases but can also be used for rapid drug and vaccine development [130].

Despite the important strides made in healthcare digitalization, there are numerous challenges to making the healthcare sector more resilient in the face of health crises. In this regard, it is necessary not only to strengthen the system but also to change its architecture toward a connected care model in which the organization, care, and assistance processes are redefined from a digital perspective and allow for making informed decisions using cutting-edge technology and BDA [22,134,135].

The transformation in health information acquisition and informed decision-making using cutting-edge technologies, however, must compromise with the mitigation of privacy associated with patient risks and data confidentiality protection [131,132]. The COVID-19 health emergency has illuminated the need for the careful consideration of the evolving relationship between privacy and public health and the relevance of the public interest in personal data processing activities [18]. These exceptional and contingent circumstances have highlighted the importance of data protection regulations and cybersecurity investment plans aimed at channeling the flow of BD into healthcare. Indeed, the collection and use of health-related data have been indispensable tools in the effort to counter and contain the pandemic [17].

Moreover, the evolution of technologies and the competitive environment require the development of new skills in the field of data analysis. In the Fourth Industrial Revolution, people continue to be the most strategic and important component in the business, and it is becoming increasingly strategic to be able to acquire analytical skills to analyze and transform consolidated data from existing fragmented data sources into valuable information for business decision-makers. In this way, it is possible to gain a competitive advantage through timely and more informed decisions based on adequate knowledge of descriptive analytics and predictive analytics, analytical techniques that are ideal for analyzing a large proportion of text-based health documents and other unstructured clinical data [135].

In conclusion, personalization of care, reduction of hospitalization, and effectiveness and cost containment of services and waiting lists are benefits unquestionably linked to digitalization and technological innovation but that require a review of the systems of traceability and control with a revolution of traditional ICT systems.

This is the challenge that healthcare must overcome. In fact, over the years, analyzing these data and sharing the results with managers and healthcare operators has made it possible to improve the level of knowledge of the system, the sustainability of the healthcare system, its accountability and transparency, and the quality and equity of care.

Our work has theoretical and practical implications. From the theoretical perspective, the paper, by proposing a literature review with a strong focus on managerial aspects, extends the literature by enriching a growing field of research. From the practical perspective, the paper reveals the need to develop new skills and redesign operational and strategic processes to consciously use heterogeneous data in future scenarios.

This paper presents several limitations. First, we used a defined set of keywords and only one database. Second, the research was conducted without programs like VOSviewer that could be used in future studies to identify new clusters related to this topic.

## Figures and Tables

**Figure 1 healthcare-10-01232-f001:**
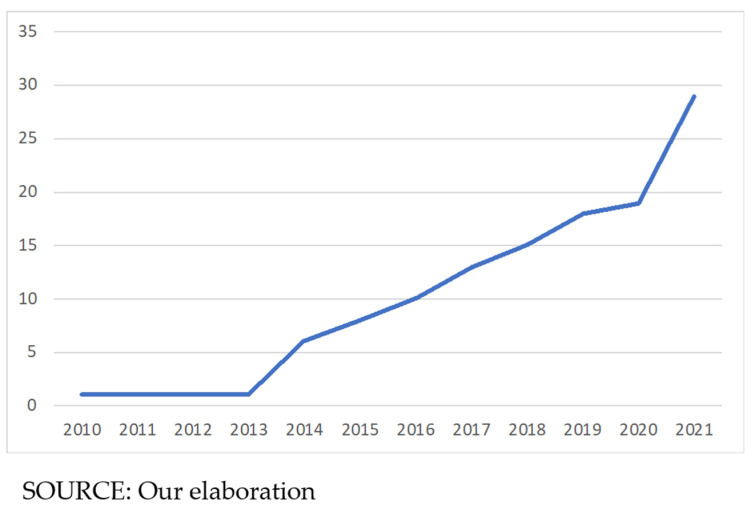
Annual distribution of publications.

**Table 1 healthcare-10-01232-t001:** Review Strategy.

Step	Selection Criteria	N. Selected Papers	N. Excluded Papers
1	Search results Scopus	305	
2	Title not relevant		27
	Record post step2	258	
3	Abstract not relevant		51
	Record post step3	206	
4	Full text non relevant		92
Record retained		135	

**Table 2 healthcare-10-01232-t002:** Top five sources.

Scientific Journal	N.
International Journal Recent Technology and Engineering	25
International Journal of Scientific &Technology Research	11
Lecture Notes in Business Information Processing	10
Big Data Research	11

**Table 3 healthcare-10-01232-t003:** Top ten articles per citations on theme 1: Big Data and awareness.

Refs.	Author	Title	Cited
[13]	Chen, H.C.; Chiang, R.H. (2012)	Business intelligence and analytics: from big data to big impact.	6862
[86]	Judd, E.; Hollander, M.D.; Brendan Carr, M. (2020)	Virtually Perfect? Telemedicine for Covid-19.	2423
[9]	Bates, D.W.; Saria, S.; Ohno-Machado, L.; Shah, A.; Escobar, G. (2014)	Big data in health care: Using analytics to identify and manage high-risk and high-cost patients	1030
[5]	Yin, Y.; Zeng, Y.; Chen, X.; Fan, Y. (2016)	The internet of things in healthcare: An overview.	592
[17]	Zhou, C.; Su, F.; Pei, T.; Zhang, A.; Du, Y.; Luo, B.; Cao, Z.; Wang, J.; Yuan, W.; Zhu, Y.; et al. (2020)	COVID-19: Challenges to GIS with Big Data.	423
[58]	Barrett, M.A.; Humblet, O. (2013)	Data and Disease Prevention: From Quantified Self to Quantified Communities	196
[72]	Srinivasan, U.; Arunasalam, B. (2013)	Leveraging big data analytics to reduce healthcare costs.	184
[38]	Gligorijević, V.; Malod-Dognin, N.; Pržulj, N. (2016)	Integrative methods for analyzing big data in precision medicine.	179
[85]	Hsieh, J.C.; Hsu, M.W. (2012)	A cloud computing based 12-lead ECG telemedicine service.	148
[57]	Hansen, M.M.; Miron-Shatz, T.; Lau, A.Y.S.; Paton, C. (2014)	Big Data in Science and Healthcare: A Review of Recent Literature and Perspectives.	133

**Table 4 healthcare-10-01232-t004:** Top ten articles per citations on theme 2: Big Data and digital transformation.

Refs.	Author	Title	Cited
[19]	Wang, Y.; Kung, L.A.; Byrd, T.A. (2018)	Big data analytics: Understanding its capabilities and potential benefits for healthcare organizations.	1134
[100]	Douglas, T.J.; Judge, Q.W.(2014)	Total Quality Management Implementation Advantage: the Role of and Competitive Structural Control and Exploration.	1116
[104]	Sharma, R.; Mithas, S.; Kankanhalli, A. (2014)	Transforming decision-making processes: A research agenda for understanding the impact of business analytics on organisations.	535
[99]	Abouelmehdi, K.; Beni-Hssane, A.; Khaloufi, H.; Saadi, M. (2017)	Big data security and privacy in healthcare: A Review.	178
[51]	Wu, J.; Li, H.; Cheng, S.; Lin, Z. (2016)	The promising future of healthcare services: When big data analytics meets wearable technology.	98
[97]	Ker, J.I.; Wang, Y.; Hajli, M.N.; Song, J.; Ker, C.W. (2014)	Deploying lean in healthcare: Evaluating information technology effectiveness in U.S. hospital pharmacies.	85
[93]	Batarseh, F.A.; Latif, E.A. (2016)	Assessing the Quality of Service Using Big Data Analytics: With Application to Healthcare.	73
[103]	Benzidia, S.; Makaoui, N.; Bentahar, O. (2021)	The impact of big data analytics and artificial intelligence on green supply chain process integration and hospital environmental performance.	63
[92]	Sheth; A.; Jaimini; U.; Thirunarayan; K.; &; Banerjee; T.; (2017)	Augmented personalized health: How smart data with IoTs and AI is about to change healthcare.	44
[107]	Wu, J.; Li, H.; Liu, L.; Zheng, H. (2017)	Adoption of big data and analytics in mobile healthcare market: An economic perspective.	37

**Table 5 healthcare-10-01232-t005:** Top ten articles per citations on theme 3: Big Data and analytical skills.

Refs.	Author	Title	Cited
[23]	Wang, Y.; Hajli, N. (2017)	Exploring the path to big data analytics success in healthcare.	340
[124]	Tambe, P. (2014)	Big data investment, skills, and firm value.	173
[120]	De Mauro, A.; Greco, M.; Grimaldi, M.; Ritala, P. (2018)	Human resources for Big Data professions: A systematic classification of job roles and required skill sets.	118
[123]	Wilder, C.R.; Ozgur, C.O. (2015)	Business Analytics Curriculum for Undergraduate Majors.	92
[62]	Sharma, P.; Sundaram, S.; Sharma, M.; Sharma, A.; Gupta, D. (2019)	Diagnosis of Parkinson’s disease using modified grey wolf optimization.	90
[72]	Wang, Y.; Kung, L.A.; Gupta, S.; Ozdemir, S. (2019)	Leveraging Big Data Analytics to Improve Quality of Care in Healthcare Organizations: A Configurational Perspective.	75
[118]	Gravili, G.; Manta, F.; Cristofaro, C.L.; Reina, R.; Toma, P. (2021)	Value that matters: intellectual capital and big data to assess performance in healthcare. An empirical analysis on the European context.	13
[119]	Tariq, M.A.; Hoyle, D.C. (2022)	Translating the Machine: Skills that Human Clinicians Must Develop in the Era of Artificial Intelligence.	3
[117]	Holm, G.R.; Lorenz, E.(2022)	The impact of artificial intelligence on skills at work in Denmark.	3
[115]	Vinay, R.; Soujanya, K.L.S.; Singh, P. (2019)	Disease prediction by using deep learning based on patient treatment history.	1

## Data Availability

Not applicable.

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
