# Peer review of "Towards the Use of Big Data in Healthcare: A Literature Review"

_healthcare, 2022, doi:10.3390/healthcare10071232_

Round 1

Reviewer 1 Report

This work aims to explore the main areas of application of Big Data (BD) in healthcare but also the restructuring of the tech-nological infrastructure and the integration of traditional data analytical tools and techniques with an elaborate computational technology able to enhance and extract information useful for decision-making. Authors also need to consider the following questions.

1. The main contribution of the paper should be highlighted and emphasized. It would be great if the drawbacks and gaps of literature are clear and, particularly, how the proposed approach aims at filling these gaps.

2. The abstract should briefly display the results of the research.

3. It is better to make a more comprehensive literature review in the form of a table (matrix) so that the reader is more confident with the contribution of this research.

4. Improve the English writing/editing of the manuscript. There are many grammatical mistakes throughout the manuscript.

5. Add more results to validate the proposed work and compared those with the existing analysis/work. Moreover, the computational effort and accuracy of the proposed work should be compared with a benchmark method and other existing work to justify its effectiveness.

6. Explain in brief how the present paper differs from the published ones.

7. What are the limitations and disadvantages of the proposed work?

8. Can the author clarify to express what contribution to the industry sector and academic field with this research?

Author Response

Dear Reviewer,

thank you for reading our contribution and for the valuable advice. We have taken them into consideration in order to improve our work, hoping to have fully understood the meaning of your suggestions.

  1. The abstract should briefly display the results of the research

We briefly displayed  and expanded the research results in the abstract.

Riga 25-30: “Our results suggest that effective and patient-centered care cannot disregard the acquisition, management, and analysis of a huge volume and variety of health data. In this way, an immediate and more effective diagnosis could be possible while maximizing healthcare resources. Deriving the benefits associated with digitization and technological innovation, however, requires the re-structuring of traditional operational and strategic processes and the acquisition of new skills.”

  1. The main contribution of the paper should be highlighted and emphasized. It would be great if the drawbacks and gaps of literature are clear and, particularly, how the proposed approach aims at filling these gaps
  2. Explain in brief how the present paper differs from the published ones

We have highlighted the main contribution work by putting more emphasis on the gaps in the literature that the study aims to fill

Riga 69-73: “Previous studies examining BD in healthcare predominantly focused on informatics [16,19,24] and medical aspects [22–24]. Few studies analyzed this topic from a managerial point of view [25,26]. Therefore, this paper contributes to this stream of literature by exploring which tangible and intangible elements are needed to draw the maximum benefits from BD.”

  1. Can the author clarify to express what contribution to the industry sector and academic field with this research?

We indicated the contribution to industry and the academic field of this research.

Riga 546-550: “Our work has theoretical and practical implications. From the theoretical perspec-tive, the paper, by proposing a literature review with a strong focus on managerial as-pects, extends the literature by enriching a growing field of research. From the practical perspective, the paper reveals the need to develop new skills and redesign operational and strategic processes to consciously use heterogeneous data in future scenarios.”.

  1. It is better to make a more comprehensive literature review in the form of a table (matrix) so that the reader is more confident with the contribution of this research

We presented the literature review in the form of tables indicated for each topic analyzed the 10 most cited studies.

Table 3 - Top ten articles per citations on theme 1: Big Data and awareness

Table 4 - Top ten articles per citations on theme 2: Big Data and digital transformation

Table 5 - Top ten articles per citations on theme 3: Big Data and analytical skills

  1. Add more results to validate the proposed work and compared those with the exiting analysis/work. Moreover, the computational effort and accuracy of the proposed work should be compared with a benchmark method and other exiting work to justify its effectiveness

We added more results to validate the work and extended the literature review to 2021.

Riga 53-59: “These data stimulate healthcare organizations to invest heavily in data analysis to facilitate decision-making [12,13]. Integrating data on an individual's unique characteristics, clinical phenotypes, and biological information obtained from diagnostic imaging to laboratory tests and medical records enables precision medicine to operate under pre-dictive and preventive conditions [14]. Having abundant data is crucial, especially in critical care environments, to be able to rapidly identify diagnoses and specific treatments for particular or rare pathological cases [15].”

Riga 126-140: “For the management and processing of these data, many healthcare sectors have adopted cloud computing. It is a solution for receiving and storing huge amounts of patient data and managing electronic medical records [46]. These heterogeneous data, when properly integrated with the most relevant health data, allow for monitoring patients' health status in various contexts (hospitals, nursing homes, private homes) [5,16,47]. This is an aspect of crucial relevance because the main errors that can lead to misdiagnosis and case fatality occur due to improper monitoring and administration of therapeutic treatment as well as to drug non-adherence [48]. Integrating data on an individual's unique characteristics, clinical phenotypes, and biological information obtained from imaging to laboratory tests and medical records enables individualized diagnostic or therapeutic solutions [14].

However, there are still many challenges to be faced. The use of cloud computing and other BD analysis tools and techniques in general, in fact, encounters a number of difficulties represented by network failures, security and privacy issues of patient data, and network downtime [46,49,50].”

Riga 153-155: “These IoT devices and sensors also play an essential role in analyzing and predicting new diseases, such as COVID-19 [55].”

Riga 160-161: “To protect patient privacy, EHR must be stored as sensitive information in a secure and reliable manner [57].”

Riga 169-194: “BD and AI technologies have high predictive capabilities in their application in the treatment of cardiovascular diseases. Studies [125,126] have shown that it is mainly machine learning techniques (k-nearest neighbor, decision tree, linear regression, and support vector machine [SVM]) that improve the accuracy of detecting heart disease.

Combinations of machine learning methods with deep learning approaches also enhance the use of neuroimaging data to classify and predict Alzheimer's disease [127]. Alzheimer's is a degenerative neurological disease that impairs a person's ability to function independently, making early diagnosis critical. Sharma et al. proposed a Ha-doop-based BD system for early indicators of the disease. Such a system involves com-bining data obtained from noninvasive magnetic resonance imaging (MRI), spectrog-raphy, magnetic resonance spectroscopy (MRS), and neuropsychological test results [128]. Kautzky et al. (2018), however, developed a prediction model based on a single diagnostic factor that allows early detection of brain abnormalities even before the onset of symptoms [129].

An additional degenerative disease in which to employ machine learning is Parkinson's disease, which affects the neurological system and limits mobility. For early detection of disease symptoms, some studies have used k-nearest neighbor, random forest, and decision tree algorithms [62,63]. Sivaparthipan et al. (2019) also highlighted the importance of data collection using cell phones to recognize the gait of Parkinson's patients [64].

Different machine learning (ML) techniques are also used to improve the prediction results for cancer, a major cause of mortality globally [133]. Torkey et al. (2021) proposed two survival prediction models based on deep learning that can guide physicians in determining breast cancer treatment options and avoid ineffective treatments [65]. In another study, Torkey et al. (2021) used an ML model that, through the construction of a DNA microarray dataset, allows for identifying discriminative features that influence the classification of different kinds of cancer and facilitate their early diagnosis [66].”

Riga 285-287: “El Samad et al. (2021), in this regard, conducted a study showing that BD management is considered a key prerogative for the quality of medical services and conditions [77].”

Riga 291-297: “Abdel-Basset et al. (2021) demonstrated the relevant role that disruptive technologies for COVID-19 analysis, such as AI, Industry 4.0, IoT, Internet of Medical Things (IoMT), BD, virtual reality (VR), drone technology and autonomous robots, 5G, and blockchain, have played in limiting the spread of COVID-19 outbreaks [81]. Another technology that is being widely deployed is Wireless Body Area Networks (WBANs). This is an innovative solution that can restructure healthcare and make pervasive support available to patients [82].”

Riga 318-321: “Fanta et al. (2021) argued that digital technologies are also a supporting tool in the healthcare sector's transition to a circular economy. Indeed, such tools, and IoT in par-ticular, can support the collection of end-of-life healthcare products as well as their recy-cling, regeneration, and disposal [91,92]”

Riga 356-358: “Managing an integrated healthcare solution requires security of medical data, which can be achieved with the cryptosystem, which has been found to be highly secure against attacks and interference [97]”

Riga 364-367: “Benzidia et al. (2021) claim that extracting new insights from existing volumes of struc-tured and unstructured data related to medical treatments and products improves deci-sion-making and enables a better understanding of each patient's costs [104].”

Riga 377-381: “Ultimately, healthcare organizations must begin to develop a concrete analysis of how to apply emerging technologies to methods such as diagnostic procedures, treatment protocol development, patient monitoring, drug development, patient diagnose, and epidemic forecasting [45]. In this way, risks can be minimized and decisions can be made from the perspective of improved effectiveness and efficiency [108,109].”

Riga 434-442: “Gravili et al. (2021) highlighted the crucial role of intangible elements, especially of the intellectual capital in the health sector. The dissemination of new knowledge and spe-cialized skills promotes the sharing of best practices in the health sector. The result is a reduction in mortality rates, better outcomes in terms of cost minimization and reduction in hospitalization periods [120,121]. De Mauro et al. (2018) proposed a classification of job roles and skill sets needed in the BD and AI era. This classification provides valuable support for business leaders and human resource managers in the selection process and in developing the skills needed to make the most of BD [122].

  1. What are the limitation and disadvantages of the proposed work?

We pointed out the limitations and disadvantages of the work.

Riga 551-553: “This paper presents several limitations. First, we used a defined set of keywords and only one database. Second, the research was conducted without programs like VOSviewer that could be used in future studies to identify new clusters related to this topic.”

  1. Improve the English writing/editing of the manuscript. There are many grammatical mistakes throughout the manuscript.

We improved the writing and editing of the manuscript in English and carried out proofreading of the work.

Reviewer 2 Report

The research needs visualization, display of data, for ease of display of information.

References in the research need to be updated 2021, 2022. Can be used

10.7717/peerj-cs.492, 10.21608/mjeer.2021.146277

The research conducted in the field can be collected using many programs and websites, like:

Connectedpapers.com , VOSviewer.

A section should be added regarding big data and disease prediction and classification in the field of healthcare

What are the areas that the research paper can add in dealing with big data and healthcare that future researchers can deal with?

Author Response

Dear Reviewer,

thank you for reading our contribution and for the valuable advice. We have taken them into consideration in order to improve our work, hoping to have fully understood the meaning of your suggestions.

  1. The research needs visualization, display of data, for ease of display of informationWe presented the literature review in the form of tables indicated for each topic analyzed the 10 most cited studies.

Table 3 - Top ten articles per citations on theme 1: Big Data and awareness

Table 4 - Top ten articles per citations on theme 2: Big Data and digital transformation

Table 5 - Top ten articles per citations on theme 3: Big Data and analytical skills

  1. References in the research need to be updated 2021, 2022. Can be used

10.7717/peerj-cs.492, 10.21608/mjeer.2021.146277

We added more results to validate the work and extended the literature review to 2021.

Riga 53-59: “These data stimulate healthcare organizations to invest heavily in data analysis to facil-itate decision-making [12,13]. Integrating data on an individual's unique characteristics, clinical phenotypes, and biological information obtained from diagnostic imaging to laboratory tests and medical records enables precision medicine to operate under pre-dictive and preventive conditions [14]. Having abundant data is crucial, especially in critical care environments, to be able to rapidly identify diagnoses and specific treatments for particular or rare pathological cases [15].”

Riga 126-140: “For the management and processing of these data, many healthcare sectors have adopted cloud computing. It is a solution for receiving and storing huge amounts of patient data and managing electronic medical records [46]. These heterogeneous data, when properly integrated with the most relevant health data, allow for monitoring patients' health status in various contexts (hospitals, nursing homes, private homes) [5,16,47]. This is an aspect of crucial relevance because the main errors that can lead to misdiagnosis and case fatality occur due to improper monitoring and administration of therapeutic treatment as well as to drug non-adherence [48]. Integrating data on an individual's unique characteristics, clinical phenotypes, and biological information obtained from imaging to laboratory tests and medical records enables individualized diagnostic or therapeutic solutions [14].

However, there are still many challenges to be faced. The use of cloud computing and other BD analysis tools and techniques in general, in fact, encounters a number of difficulties represented by network failures, security and privacy issues of patient data, and network downtime [46,49,50].”

Riga 153-155: “These IoT devices and sensors also play an essential role in analyzing and predicting new diseases, such as COVID-19 [55].”

Riga 160-161: “To protect patient privacy, EHR must be stored as sensitive information in a secure and reliable manner [57].”

Riga 285-287: “El Samad et al. (2021), in this regard, conducted a study showing that BD management is considered a key prerogative for the quality of medical services and conditions [77].”

Riga 291-297: “Abdel-Basset et al. (2021) demonstrated the relevant role that disruptive technologies for COVID-19 analysis, such as AI, Industry 4.0, IoT, Internet of Medical Things (IoMT), BD, virtual reality (VR), drone technology and autonomous robots, 5G, and blockchain, have played in limiting the spread of COVID-19 outbreaks [81]. Another technology that is being widely deployed is Wireless Body Area Networks (WBANs). This is an innovative solution that can restructure healthcare and make pervasive support available to patients [82].”

Riga 318-321: “Fanta et al. (2021) argued that digital technologies are also a supporting tool in the healthcare sector's transition to a circular economy. Indeed, such tools, and IoT in par-ticular, can support the collection of end-of-life healthcare products as well as their recy-cling, regeneration, and disposal [91,92]”

Riga 356-358: “Managing an integrated healthcare solution requires security of medical data, which can be achieved with the cryptosystem, which has been found to be highly secure against attacks and interference [97]”

Riga 364-367: “Benzidia et al. (2021) claim that extracting new insights from existing volumes of struc-tured and unstructured data related to medical treatments and products improves deci-sion-making and enables a better understanding of each patient's costs [104].”

Riga 377-381: “Ultimately, healthcare organizations must begin to develop a concrete analysis of how to apply emerging technologies to methods such as diagnostic procedures, treatment protocol development, patient monitoring, drug development, patient diagnose, and epidemic forecasting [45]. In this way, risks can be minimized and decisions can be made from the perspective of improved effectiveness and efficiency [108,109].”

Riga 434-442: “Gravili et al. (2021) highlighted the crucial role of intangible elements, especially of the intellectual capital in the health sector. The dissemination of new knowledge and spe-cialized skills promotes the sharing of best practices in the health sector. The result is a reduction in mortality rates, better outcomes in terms of cost minimization and reduction in hospitalization periods [120,121]. De Mauro et al. (2018) proposed a classification of job roles and skill sets needed in the BD and AI era. This classification provides valuable support for business leaders and human resource managers in the selection process and in developing the skills needed to make the most of BD [122].

  1. The research conducted in the field can be collected using many programs and websites, like:

Connectedpapers.com , VOSviewer.

We pointed out the limitations and disadvantages of the work.

Riga 551-553: “This paper presents several limitations. First, we used a defined set of keywords and only one database. Second, the research was conducted without programs like VOSviewer that could be used in future studies to identify new clusters related to this topic.”

  1. A section should be added regarding big data and disease prediction and classification in the field of healthcare

We have added a section on big data and disease prediction and classification in health care. 

 Riga 169-194: “BD and AI technologies have high predictive capabilities in their application in the treatment of cardiovascular diseases. Studies [125,126] have shown that it is mainly machine learning techniques (k-nearest neighbor, decision tree, linear regression, and support vector machine [SVM]) that improve the accuracy of detecting heart disease.

Combinations of machine learning methods with deep learning approaches also enhance the use of neuroimaging data to classify and predict Alzheimer's disease [127]. Alzheimer's is a degenerative neurological disease that impairs a person's ability to function independently, making early diagnosis critical. Sharma et al. proposed a Ha-doop-based BD system for early indicators of the disease. Such a system involves com-bining data obtained from noninvasive magnetic resonance imaging (MRI), spectrog-raphy, magnetic resonance spectroscopy (MRS), and neuropsychological test results [128]. Kautzky et al. (2018), however, developed a prediction model based on a single diagnostic factor that allows early detection of brain abnormalities even before the onset of symptoms [129].

An additional degenerative disease in which to employ machine learning is Parkinson's disease, which affects the neurological system and limits mobility. For early detection of disease symptoms, some studies have used k-nearest neighbor, random forest, and decision tree algorithms [62,63]. Sivaparthipan et al. (2019) also highlighted the importance of data collection using cell phones to recognize the gait of Parkinson's patients [64].

Different machine learning (ML) techniques are also used to improve the prediction results for cancer, a major cause of mortality globally [133]. Torkey et al. (2021) proposed two survival prediction models based on deep learning that can guide physicians in determining breast cancer treatment options and avoid ineffective treatments [65]. In another study, Torkey et al. (2021) used an ML model that, through the construction of a DNA microarray dataset, allows for identifying discriminative features that influence the classification of different kinds of cancer and facilitate their early diagnosis [66].”

  1. What are the areas that the research paper can add in dealing with big data and healthcare that future researchers can deal with?

We indicated the contribution to industry and the academic field of this research and future research line.

Riga 546-550: “Our work has theoretical and practical implications. From the theoretical perspec-tive, the paper, by proposing a literature review with a strong focus on managerial as-pects, extends the literature by enriching a growing field of research. From the practical perspective, the paper reveals the need to develop new skills and redesign operational and strategic processes to consciously use heterogeneous data in future scenarios.”.

Riga 551-553: “This paper presents several limitations. First, we used a defined set of keywords and only one database. Second, the research was conducted without programs like VOSviewer that could be used in future studies to identify new clusters related to this topic.”

Reviewer 3 Report

Grazia et al. conducted a comprehensive literature review about how "big data" and "artificial intelligence" are incorporated into the field of healthcare. The literature review strategy is well elaborated. The connection between health awareness, digital transformation, analytical skills and big data is the major focus of the discussion.

Overall, the manuscript provides a systematic review of literatures that associate big data with healthcare. Some formatting related minor issues do exist throughout the manuscript, which may require a good amount of editing. 

Author Response

Dear Reviewer,

thank you for reading our contribution and for the valuable advice. We have taken them into consideration in order to improve our work, hoping to have fully understood the meaning of your suggestions.

  1. Some formatting related minor issues do exist throughout the manuscript, which may require a good amount of editing

We improved the writing and editing of the manuscript in English and carried out proofreading of the work.

Round 2

Reviewer 1 Report

The authors have revised and responded to the suggestions. I think it's acceptable.

Reviewer 2 Report

Accept in present form